# Biologically Active Echinulin-Related Indolediketopiperazines from the Marine Sediment-Derived Fungus *Aspergillus niveoglaucus*

**DOI:** 10.3390/molecules25010061

**Published:** 2019-12-23

**Authors:** Olga F. Smetanina, Anton N. Yurchenko, Elena V. Girich (Ivanets), Phan Thi Hoai Trinh, Alexander S. Antonov, Sergey A. Dyshlovoy, Gunhild von Amsberg, Natalya Y. Kim, Ekaterina A. Chingizova, Evgeny A. Pislyagin, Ekaterina S. Menchinskaya, Ekaterina A. Yurchenko, Tran Thi Thanh Van, Shamil S. Afiyatullov

**Affiliations:** 1G.B. Elyakov Pacific Institute of Bioorganic Chemistry, Far Eastern Branch of the Russian Academy of Sciences, Prospect 100-letiya Vladivostoka, 159, Vladivostok 690022, Russia; smetof@rambler.ru (O.F.S.); alexanderantonovpiboc@gmail.com (A.S.A.); dyshlovoy@gmail.com (S.A.D.); natalya_kim@mail.ru (N.Y.K.); martyyas@mail.ru (E.A.C.); pislyagin@hotmail.com (E.A.P.); ekaterinamenchinskaya@gmail.com (E.S.M.); dminae@mail.ru (E.A.Y.); afiyat@piboc.dvo.ru (S.S.A.); 2Department of Marine Biotechnology, Nhatrang Institute of Technology Research and Application, Vietnam Academy of Science and Technology, 650000 Nha Trang, Vietnam; phanhoaitrinh84@gmail.com (P.T.H.T.); tranthanhvan@nitra.vast.vn (T.T.T.V.); 3School of Natural Science, Far Eastern Federal University, Sukhanova St., 8, Vladivostok 690000, Russia; 4Laboratory of Experimental Oncology, Department of Oncology, Hematology and Bone Marrow Transplantation with Section Pneumology, Hubertus Wald-Tumorzentrum, University Medical Center Hamburg-Eppendorf, 20246 Hamburg, Germany; g.von-amsberg@uke.de; 5Martini-Klinik Prostate Cancer Center, University Hospital Hamburg-Eppendorf, 20246 Hamburg, Germany

**Keywords:** *Aspergillus niveoglaucus*, echinulin, neoechinulin, cryptoechinuline, NMR data, neuroprotective activity, 6-OHDA, paraquat, rotenone, urease inhibition, cytotoxic activity

## Abstract

Seven known echinulin-related indolediketopiperazine alkaloids (**1**–**7**) were isolated from the Vietnamese sediment-derived fungus *Aspergillus niveoglaucus*. Using chiral HPLC, the enantiomers of cryptoechinuline B (**1**) were isolated as individual compounds for the first time. (+)-Cryptoechinuline B (**1a**) exhibited neuroprotective activity in 6-OHDA-, paraquat-, and rotenone-induced in vitro models of Parkinson’s disease. (−)-Cryptoechinuline B (**1b**) and neoechinulin C (**5**) protected the neuronal cells against paraquat-induced damage in a Parkinson’s disease model. Neoechinulin B (**4**) exhibited cytoprotective activity in a rotenone-induced model, and neoechinulin (**7**) showed activity in the 6-OHDA-induced model.

## 1. Introduction

Many fungi belonging to the genera *Aspergillus* and *Eurotium* produce echinulin-related indole diketopiperazine alkaloids [1]. Currently, the class of echinulin congeners includes more than 50 members. These alkaloids originate from the dipeptide cyclo-l-alanyl-l-tryptophan via prenylation by two types of prenyltransferases, which ultimately results in the reverse prenyl group at C-2 and normal prenyl groups at C-4–C-7 on the indole moiety [2]. Another diverse group of metabolites that are commonly produced by *Aspergillus* and *Eurotium* are auroglaucin-related polyketides [3]. To date, only cryptoechinulines B and D and two other metabolites [4,5] are known to be mixed metabolic products of echinulin-related indolediketopiperazines and auroglaucin-related polyketides [4,5]. Thus, they were very likely formed via a Diels-Alder reaction between auroglaucin and neoechinulin B or neoechinulin C. This assumption was confirmed by the synthesis of cryptoechinuline D [6]. These metabolites have been isolated as racemic mixtures, and recently, cryptoechinuline D was separated into its individual enantiomers using chiral HPLC [5].

Echinulin-related compounds exhibit diverse biological activities, such as antimicrobial [7], cytotoxic [8], antiviral [8] and antioxidant [9]. Interestingly, neoechinulin A, one of the most studied echinulin-related alkaloids, was reported to have neuroprotective activity [10]. Neoechinulin A protects PC12 cells from damage induced by MPP+ [11], rotenone [12], superoxide (O^2−^), and NO cogenerator (3-(4-morpholinyl)sydnonimine hydrochloride) (SIN-1) [10,13]. Although the structure-activity relationship of neoechinulin A was investigated, those of other natural echinulin-related compounds have not been studied.

In the present study, we investigated the extract of the fungus *Aspergillus niveoglaucus* (=*Eurotium niveoglaucum*) 01NT.1.10.4, a known producer of auroglaucin-related polyketides [14], isolated from a sediment sample collected in Nha Trang Bay (Vietnam). Here, we report the isolation and identification of echinulin-related indolediketopiperazines (Figure 1) from this marine fungus and the structure-activity relationship of these compounds focusing mainly on their neuroprotective activities.

## 2. Results

### 2.1. Isolation and Identification of Compounds ***1***–***7***

The HRESIMS spectrum of compound **1** suggested a molecular formula of C_43_H_49_N_3_O_5_ (calcd for C_43_H_49_N_3_NaO_5_, 710.3564), which corresponds to 21 double-bond equivalents. An analysis of the ^1^H and ^13^C-NMR data together with the DEPT and HSQC data (Appendix A) led to the identification of **1** as a known compound, cryptoechinuline B [15]. A direct comparison of the NMR data of **1** with literature data [15,16] confirmed the full identity of the planar structure of **1**. The optical rotation of **1** was near zero, indicating that **1** was a racemic mixture. Using chiral HPLC, enantiomers **1a** and **1b** were separated. A comparison of their CD data (Figure 2, Appendix A) and optical rotations with reported values indicated absolute stereochemistries of 12*R*,26*S*,29*R* and 12*S*,26*R*,29*S* for (+)- and (−)-cryptoechinulines D, **1a** and **1b**, respectively (Figure 1). We also identified cryptoechinuline D (**2**); echinulin (**3**); neoechinulins B (**4**), C (**5**), and E (**6**), and neoechinulin (**7**).

### 2.2. Biological Activities of the Isolated Compounds

The neuroprotective activities of the compounds in the toxin-induced models of Parkinson’s disease (PD) were investigated in murine malignant Neuro-2a cells because various neuroblastoma cell lines are widely used as in vitro models for Parkinson’s disease (PD) [17,18,19].

We first evaluated the cytotoxicities of **1**–**7** and found that **1a**, **1b**, **2**, **3**, **6**, and **7** were noncytotoxic to Neuro-2a cells at concentrations up to 100 µM (Appendix A), while **4** and **5** showed moderate cytotoxicities with IC_50_ values of 50.9 and 40.6 µM, respectively (Appendix A). Note that **4** was reported to be cytotoxic against murine macrophage RAW264.7 cells with a potency similar to that reported here [20]. Our results support the idea that the α,β-unsaturated carbonyl moiety in **4** and **5** control the toxicity of these molecules [21]. Of note, compounds **3** and **7** were moderately cytotoxic against human drug-resistant and drug-sensitive prostate cancer cells (Appendix A). Echinulin (**3**) exhibited moderate cytotoxicity towards 22Rv1, PC-3, and LNCaP cells with IC_50_ values of 63.2, 41.7, and 25.9 µM, respectively. Neoechinulin (**7**) showed similar activities with IC_50_ values of 49.9, 63.8, and 38.9 µM, respectively.

We next investigated the antioxidant properties of **1**–**7** in cell-based assays and cell-free assays.

Oxidative stress is a key factor in Parkinson’s disease. Reactive oxygen species cause serious damage and death of dopamine-producing cells when the antioxidant capacity of the cell is reduced against oxidative stress [22,23]. For this reason, compounds may be very effective in treating PD based on in vitro and in vivo models if they demonstrated free radical scavenger effects in cell-free assays or anti-ROS influences in cell-based tests [24].

Compounds **1a** and **1b** suppressed paraquate (PQ)-induced upregulation of intracellular ROS (Table 1). However, these substances did not show any effect on ROS levels in the 6-OHDA-induced PD model (Table 1). In addition, both **1a** and **1b** exhibited weak radical scavenging activities towards DPPH free radicals (Table 1). Cryptoechinuline D (**2**), which was previously reported to have strong DPPH-radical scavenging activity [25], also demonstrated antioxidant effects in a PQ-induced Neuro-2a cell model, where pretreatment with **2** could suppress ROS upregulation (Table 1). Compounds 3-6 had activities similar to those of 2 in both the PQ and DPPH [25,26] assays (Table 1), whereas the radical-scavenging properties of neoechinulin (**7**) have not been previously investigated. Thus, we showed for the first time that neoechinulin (**7**) binds DPPH free radicals with an IC_50_ of 62.6 µM (Table 1). Moreover, **7** was active both in PQ- and 6-OHDA-treated cell models.

Compounds **1a** and **1b** exhibited neuroprotective activities in toxin-induced PD models using Neuro-2a cells. In the PQ-treated cell model, **1a** and **1b** increased cell viability by 21.6% and 54.4%, respectively (Figure 3). In the 6-OHDA-treated model, however, **1a** increased the cell viability by 40.7%, while **1b** was inactive. **1a** and **1b** showed similar effects in the rotenone-induced model, i.e., **1a** increased the cell viability by 79.6%, but **1b** was inactive (Figure 3). Cryptoechinuline D (**2**) did not show any cytoprotective effects in any of the cell-based models tested (Figure 3).

Neoechinulin B (**4**) and neoechinulin E (**6**) induced significant increases in the viability of rotenone-treated cells by 68.4% and 55.6%, respectively, whereas **5** and **7** were not active (Figure 3). Neoechinulins C (**5**) and E (**6**) increased the viability of PQ-treated cells by 27% and 28%, respectively, while echinulin (**3**), neoechinulin B (**4**), and neoechinulin (**7**) were inactive. Compounds **3**–**7** did not demonstrate cytoprotective effects in 6-OHDA-treated Neuro-2a cells (Figure 3).

## 3. Discussion

Parkinson’s disease is one of the most common age-related motoric neurodegenerative diseases. The pathogenesis of PD includes neuronal death as a result of oxidative stress mediated by increasing intracellular levels of reactive oxygen species (ROS) and reactive nitrogen species. The hyperproduction of ROS results in damage to cell components such as DNA, lipids, and proteins. The peroxidation of the latter promotes mitochondrial injury. In addition, ROS cause mitochondrial dysfunction as well as activation of apoptosis-related death signaling, resulting in neuronal cell death. Thus, compounds with antioxidant properties may have therapeutic potential as PD-preventive agents [27].

The neurotoxins 6-hydroxydopamine, paraquat, and rotenone, ROS inducers, are commonly used for investigations of neuroprotective effects in PD-like in vitro and in vivo models [18,28]. However, they induce ROS formation through different mechanisms. Thus, rotenone is reported to act as a complex I inhibitor, and it is expected that mitochondrial dysfunction results in superoxide anion formation. 6-OHDA has been indicated to produce ROS through enzymatic or nonenzymatic auto-oxidation. In the case of paraquat, ROS are mainly generated via redox cycling [29]. To model these processes, various cell lines, such as human neuroblastoma SHSY5Y [30] and mouse neuroblastoma Neuro-2a [31,32] as well as primary dopaminergic neurons [33], human neuroglioma H4 cells [34], and others, are often used.

To date, only a few marine fungal metabolites are known to be neuroprotective [24]. Neoechinulin A is well known for its antioxidant properties [35]. Its neuroprotective and anti-inflammatory effects, as well as structure-activity relationships, have been investigated in detail [10,20,36,37]. In this study, we isolated seven previously known echinulin-related compounds from the Vietnamese sediment-derived fungus *Aspergillus niveoglaucus* and investigated their neuroprotective properties.

We successfully separated enantiomers (+)- and (−)-cryptoechinulines B and compared their biological activities. (+)-Cryptoechinuline B (**1a**) induced a significant increase in the viability of cells incubated with all the neurotoxins used (i.e., rotenone, paraquat, and 6-OHDA), while this effect was limited in the paraquat model for (−)-cryptoechinuline B (**1b**).

Both stereoisomers of cryptoechinuline B suppress ROS upregulation and increase cell viability in the paraquat-induced model. It can be assumed that the neuroprotective effects, in this case, are due to the antioxidant effects of both stereoisomers. In the models induced by 6-OHDA and rotenone, however, another neuroprotective pathway that is not associated with the antioxidant but depends on the stereochemistry of the molecules is likely active. This suggestion corroborates the reports that paraquat, rotenone, and 6-OHDA induce ROS formation through different mechanisms [18]. Importantly, the different metabolic changes induced by the neurotoxins were not associated with differences in the levels of ROS [29].

Enantiomers having different biological activities is well-known and widespread [38]. This is also known for the drugs currently used for PD treatment, e.g., L-DOPA. Generally, the stereoselectivity of the ligands affects their interactions with a specific target protein or drug metabolism [39]. The differences between the neuroprotective effects of (+)- and (−)-cryptoechinulines B in 6-OHDA- and rotenone-induced models arose from their interactions with the target proteins, although the identification of these targets may require additional experiments. It was reported that chromogranin B (CHGB) is a plausible target of neoechinulin A, as it binds to CHGB and modulates its functions to exert its cytoprotective activity [37], whereas, for the enantiomers of cryptoechinuline B, their targets require further study.

It was recently shown that fungal metabolites may exhibit neuroprotective activities in one or both 6-OHDA- and paraquat-induced Parkinson’s disease models [40]. In the present work, **4**–**6** showed different activity profiles in the three neurotoxin-induced cell models.

Neoechinulin A can significantly delay rotenone-induced death in PC12 cells. The presence of the C8/C9 double bond in neoechinulin A is thought to be an essential structural element required for the cytoprotective effect in the rotenone-induced model [12]. In our case, the fact that **1a**, **4**, and **6**, which are C8/C9 double bond congeners, induced significant increases in the viability of rotenone-treated cells, while saturated echinulin (**3**) was inactive. This suggests the essential role of the C8/C9 double bond, which forms a conjugate system with the indole and diketopiperazine moieties of neoechinulins, in the neuroprotective activity of the echinulin-related compounds. A comparison of the protective effects of compounds **4**–**7** against rotenone suggests that nonprenylated compounds (similar to neoechinulin A) are more effective than compounds having prenyl groups on the indole moiety.

## 4. Materials and Methods

### 4.1. General

Optical rotations were measured on a Perkin-Elmer 343 polarimeter (Perkin Elmer, Waltham, MA, USA). UV spectra were recorded on a Specord UV VIS spectrometer (Carl Zeiss, Jena, Germany) in MeOH. CD spectra were measured with a Chirascan-Plus CD spectrometer (Applied Photophysics, Leatherhead, United Kingdom) in MeOH. NMR spectra were recorded in DMSO-d_6_ with Bruker DPX-500 (Bruker BioSpin GmbH, Rheinstetten, Germany) or Bruker DRX-700 (Bruker BioSpin GmbH, Rheinstetten, Germany) spectrometers using TMS as an internal standard. HRESIMS spectra were measured on a Maxis impact mass spectrometer (Bruker Daltonics GmbH, Rheinstetten, Germany).

Low-pressure liquid column chromatography was performed using silica gel (50/100 μm, Imid, Russia). Plates (4.5 × 6.0 cm) precoated with silica gel (5–17 μm, Imid) and silica gel 60 RP-18 F_254_S (20 × 20 cm, Merck KGaA, Germany) were used for thin-layer chromatography. Preparative HPLC was carried out with a Shimadzu LC-20 chromatograph (Shimadzu USA Manufacturing, Canby, OR, USA) using YMC ODS-AM (YMC Co., Ishikawa, Japan) (5 µm, 10 mm × 250 mm) and YMC SIL (YMC Co., Ishikawa, Japan) (5 µm, 10 mm × 250 mm) columns with a Shimadzu RID-20A refractometer (Shimadzu Corporation, Kyoto, Japan) and with an Agilent 1100 chromatograph (Agilent Technologies, San Jose, CA, USA) using a Kromasil column (Nouryon, Bohus, Sweden) (5 µm, 4.6 mm × 150 mm) with an Agilent 1100 refractometer (Agilent Technologies, San Jose, CA, USA).

### 4.2. Fungal Strain

The strain of *A. niveoglaucus* was isolated from a marine sediment sample (Nha Trang Bay, South China Sea, Vietnam) and identified as described earlier [14]. The strain is stored at the collection of microorganisms of the Nha Trang Institute of Technology and Research Application VAST (Nha Trang, Vietnam) under the code 01NT.1.10.4.

### 4.3. Cultivation of the Fungus

The fungus was grown without shaking at 28 °C for three weeks in 40 × 500 mL Erlenmeyer flasks each containing rice (20.0 g), yeast extract (20.0 mg), KH_2_PO_4_ (10 mg), and natural seawater from Nha Trang Bay (40 mL).

### 4.4. Extraction and Isolation

The fungal mycelia with the medium were extracted for 24 h with 12.0 L of EtOAc. Evaporation of the solvent under reduced pressure gave a dark brown oil (3.0 g). To this residue was added 250 mL of H_2_O-EtOH (4:1), and the mixture was thoroughly stirred to yield a suspension. The suspension was sequentially extracted with hexane (150 mL × 2), EtOAc (150 mL × 2) and n-BuOH (150 mL × 2). During this extraction, crystals of **3** (475.5 mg) were obtained. The EtOAc fraction was concentrated in vacuo to give a dry residue (1.9 g), which was separated on a silica gel column (35.0 × 2.5 cm) eluted with a hexane-EtOAc gradient (1:0–0:1). The hexane-EtOAc fraction EN-1-19 (85:15, 73 mg) was purified using Sephadex LH-20 eluting with chloroform to yield **5** (4.0 mg) and fraction EN-5-7 (30.0 mg). Fraction EN-5-7 was purified by RP-HPLC on a YMC ODS-AM column eluting with MeOH-H_2_O (95:5) to yield **1** (6.1 mg) as a mixture of enantiomers. The mixture of **1** was separated by HPLC on a Kromasil chiral column eluting with MeOH-H_2_O-TFA (95:5:0.1) to yield **1a** (0.57 mg) and **1b** (0.6 mg). The hexane-EtOAc fraction EN-1-20 (85:15, 150 mg) was separated using Sephadex LH-20 eluting with chloroform and subsequent HPLC separations on a YMC SIL column (CHCl_3_-hexane, 80:20) yielded **2** (11.0 mg) as a mixture of enantiomers and on a YMC ODS-AM column (MeOH-H_2_O, 90:10) yielded **4** (7.6 mg). The hexane-EtOAc fraction EN-1-38 (80:20, 250 mg) was purified using Sephadex LH-20 eluting with chloroform to yield **6** (15.0 mg) and fraction EN-13-14 (22 mg), which was purified by HPLC on a YMC ODS-AM column eluting with MeOH-H_2_O (70:30) to yield **7** (10.2 mg).

*(+)-Cryptoechinuline B* (**1a**): White powder; ${\left[\alpha \right]}_{D}^{20}$ + 191.6° (c 0.05, MeOH); UV (MeOH) λ_max_ (logε) 352 (3.77), 280 (4.01), 229 (4.56), 198 (4.55); CD (0.21 mM, MeOH) λ_max_ (Δε) 195 (+0.43), 205 (+0.61), 230 (−0.87), 250 (+2.59), 280 (+1.03), 300 (+0.59), 350 (-0.06), 390 (+0.81), 450 (+0.05); ^1^H and ^13^C-NMR data, see Appendix A; HRESIMS [M + Na]^+^ 710.3550 (calcd for C_43_H_49_N_3_NaO_5_, 710.3564).

*(−)-Cryptoechinuline B* (**1b**): White powder; ${\left[\alpha \right]}_{D}^{20}$ − 190° (c 0.05, MeOH); UV (MeOH) λ_max_ (logε) 352 (3.77), 280 (4.01), 229 (4.56), 198 (4.55); CD (0.21 mM, MeOH) λ_max_ (Δε) 195 (−1.84), 205 (−0.50), 230 (+1.08), 250 (−3.83), 280 (−1.45), 300 (−1.39), 350 (+0.50), 390 (−1.61), 450 (−0.13); ^1^H and ^13^C-NMR data, see Appendix A; HRESIMS [M + Na]^+^ 710.3550 (calcd for C_43_H_49_N_3_NaO_5_, 710.3564).

### 4.5. DPPH-Radical Scavenger Assay

The DPPH radical scavenging activities of the compounds were tested as described [41].

The compounds were dissolved in MeOH, and the solutions (120 µL) were dispensed into wells of a 96-well microplate. In all, 30 µL of the DPPH (Sigma-Aldrich, Steinheim, Germany) solution in MeOH (7.5 × 10^−3^ M) was added to each well. The concentrations of the test compounds in the mixtures were 10 and 100 µM. The mixtures were shaken and left to stand for 30 min, and the absorbance of the resulting solutions was measured at 520 nm with a microplate reader MultiscanFC (ThermoScientific, Waltham, MA, USA). The radical scavenging activities of the compounds at 100 µM are presented as % relative to the control (MeOH alone), and the concentration scavenging 50% of the DPPH radical (EC_50_) was calculated for each compound.

### 4.6. Bioassays

#### 4.6.1. Cell Culture

The human prostate cancer cell lines 22Rv1, PC-3, and LNCaP and the murine neuroblastoma cell line Neuro-2a were purchased from ATCC.

22Rv1, PC-3, and LNCaP cell lines were cultured according to the manufacturer’s instructions in 10% FBS/RPMI media (Invitrogen, Carlsbad, CA, USA). Cells were continuously kept in culture for a maximum of 3 months and were routinely inspected microscopically for stable phenotypes and regularly checked for contamination with mycoplasma. Cell line authentication was performed by DSMZ (Braunschweig, Germany) using highly polymorphic short tandem repeat loci [42].

Neuro-2a cells were cultured in DMEM medium containing 10% fetal bovine serum (Biolot, St. Petersburg, Russia) and 1% penicillin/streptomycin (Invitrogen, Carlsbad, CA, USA). Cells were incubated at 37 °C in a humidified atmosphere containing 5% (*v*/*v*) CO_2_ [43].

#### 4.6.2. Cytotoxicity Assay

The in vitro cytotoxicities of the individual substances were evaluated using an MTT (3-(4,5-dimethylthiazol-2-yl)-2,5-diphenyltetrazolium bromide) assay, which was performed according to the manufacturer’s instructions (Sigma-Aldrich, St. Louis, MO, USA). The results are presented as viability as a % of the control, and the concentration inhibiting cell viability by 50% (IC_50_) was calculated. Docetaxel was used as a reference substance.

#### 4.6.3. Neurotoxin-Induced Cell Models of Parkinson’s Disease

Neuroblastoma Neuro-2a line cells (1 × 10^4^ cells/well) were treated with the test compounds at concentrations of 10 µM for 1 h, and then the neurotoxins at different concentrations were added to the neuroblastoma cell suspensions [40]. Rotenone (Sigma-Aldrich, USA) was used at 10 µM. Paraquat (Sigma-Aldrich, USA) was used at 500 µM. 6-Hydroxydopamine (Sigma-Aldrich, USA) was used at 50 µM. Cells incubated without neurotoxins and the test compounds and cells incubated with neurotoxins only were used as positive and negative controls, respectively. After 24 h of incubation, the cell viabilities were measured using the MTT method. The results are presented as the viability % relative to the control.

#### 4.6.4. Reactive Oxygen Species Levels in Neurotoxins-Treated Cells

Cell suspensions (1×10^4^ cells/well) were incubated with solutions of the test compound (10 µM) for 1 h. Then, 6-hydroxydopamine solution (Sigma-Aldrich, 50 µM) or paraquat solution (Sigma-Aldrich, 500 µM) were added to each well. After 30 min (for 6-hydroxydopamine) or 1 h (for paraquat), 20 μL of 2,7-dichlorodihydrofluorescein diacetate (H_2_DCF-DA) solution (Molecular Probes, final concentration 10 mM) was added to each well, and the microplate was incubated for an additional 10 min at 37 °C. The intensity of dichlorofluorescin fluorescence was measured at λ_ex_ = 485 nm and λ_em_ = 518 nm. The results are presented as % relative to the control [44].

## 5. Conclusions

The individual enantiomers (+)- and (−)-cryptoechinuline B were isolated for the first time using chiral HPLC. This is the first report of the antioxidant and neuroprotective activities of (+)- and (−)-cryptoechinuline B. The neuroprotective activities of (+)-cryptoechinuline B (**1a**) and (−)-cryptoechinuline B (**1b**) towards 6-OHDA and rotenone models were different due to their different absolute configurations. Thus, (+)-cryptoechinuline B (**1a**) exhibited neuroprotective effects in all neurotoxin-induced in vitro models of PD and may be a promising lead compound.

## Figures and Tables

**Figure 1 molecules-25-00061-f001:**
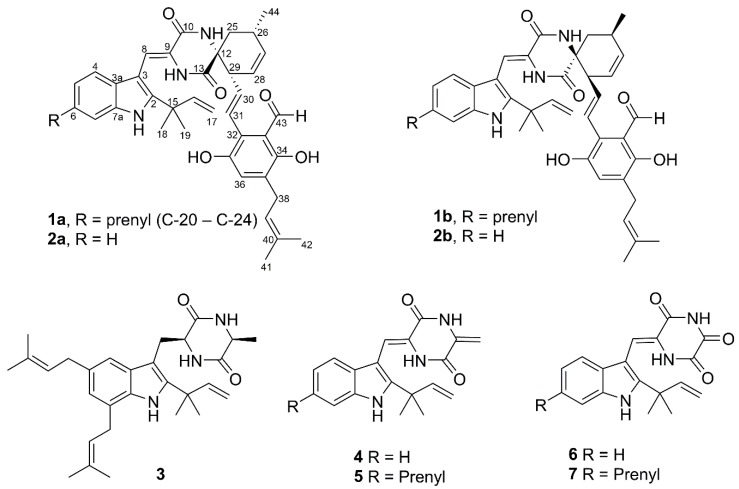
The structures of isolated compounds **1**–**7**.

**Figure 2 molecules-25-00061-f002:**
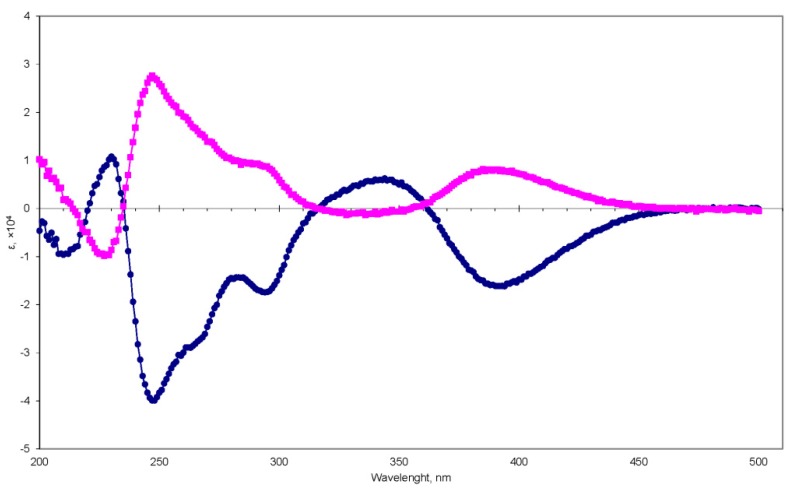
CD curves of **1a** (magenta, ■) and **1b** (blue, ●).

**Figure 3 molecules-25-00061-f003:**
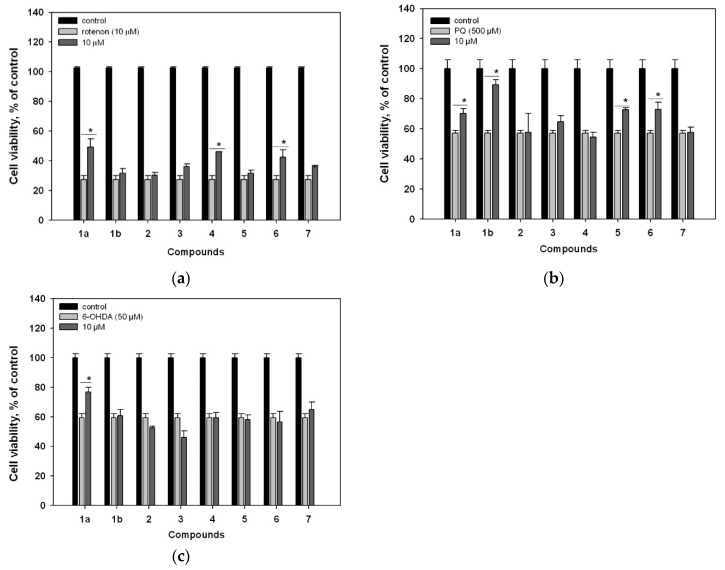
Effects of compounds **1**–**7** on the viability of Neuro-2a cells treated with rotenone (**a**), PQ (**b**), and 6-OHDA (**c**). Cells were pretreated with 10 µM of the tested compound (**1**–**7**) for 1 h, and then the neurotoxin was added. * Statistically significant difference (*p* ≤ 0.05, Student’s t-test).

**Table 1 molecules-25-00061-t001:** Antioxidant activities of compounds **1**–**7** in cell-based and cell-free assays.

Compounds	ROS Level, % of Control		DPPH Scavenger Activity IC_50_, µM
	Paraquat (PQ)	6-OHDA	
**neurotoxin**	148.4. ± 6.9	139.2 ± 1.8	-
**1a**	94.1 ± 4.1	132.9 ± 2.7	122.4 ± 7.1
**1b**	83.2 ± 2.3	121.2 ± 1.6	118.1 ± 4.7
**2**	92.8 ± 2.4	-	23.6 [25]
**3**	90.3 ± 2.2	-	29.9 [26]
**4**	101.8 ± 7.6	-	34.0 [26]
**5**	97.4 ± 2.8	-	31.1 [26]
**6**	94.1 ± 2.9	-	46.0 [25]
**7**	101.8 ± 7.6	95.4 ± 3.4	62.6 ± 1.1

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
