# Peer review of "Biologically Active Echinulin-Related Indolediketopiperazines from the Marine Sediment-Derived Fungus Aspergillus niveoglaucus"

_molecules, 2019, doi:10.3390/molecules25010061_

Round 1
Reviewer 1 Report
The current manuscript reports chiral isolation of echinulins with their neuroprotective activities.
Manuscript prepared well and results are clear. However, there are some points require the attention of the authors as below.
1. In figure 1, the absolute structure of C-12 compounds is different from reference 8 and Organic & Biomolecular Chemistry, 2012, 10, 9501. Please check it.
2. Diketopiperazines are notorious with cytotoxicity. Despite the results on table S1 and short comment in Conclusion, this point should be addressed more in the manuscript such as Discussion.
Author Response
Dear reviewer, thank you for appreciation of our study and the thorough analysis of our manuscript. Below you can find the answers for your comments.
Q: 1. In figure 1, the absolute structure of C-12 compounds is different from reference 8 and Organic & Biomolecular Chemistry, 2012, 10, 9501. Please check it.
A: The absolute configurations of all stereocenters in 1a and 1b are same with Org. Biomol. Chem., 2012, 10, 9501. Unfortunately, the ref. 8 contains the error in the configurations of cryproechinuline D and we have made a few mistakes too. Therefore, we have revised following points:
figure 1, structures of 1a and 1b, the absolute configurations were assigned for 1a and 1b as 12R, 26S, 29R and 12S, 26R, 29S, respectively (instead of 12R, 26S, 29S and 12S, 26R, 29R, respectively), the figure 2 caption was revised
Q: 2. Diketopiperazines are notorious with cytotoxicity. Despite the results on table S1 and short comment in Conclusion, this point should be addressed more in the manuscript such as Discussion.
A: This study has not been focused on cytotoxicity investigation due to non-outstanding IC50 values for isolated compounds. These data corresponding with literature data for these and related compounds.
Reviewer 2 Report
This manuscript that describes the isolation of echulin diketopiperazines from a marine Aspergillus species. The natural products were tested for biological activity and displayed interesting neuroprotective activity.
Overall, the manuscript is clearly written and the results will be of interest to the natural products community.
Author Response
Dear reviewer, thank you for appreciation of our study.
Reviewer 3 Report
In the manuscript, Smetanina describe the isolation of a series of indole diketopiperazines from an EtOAc extract obtained from cultures of a marine strain of Aspergillus niveoglaucus, as well as the in vitro neuroprotective effects in Parkinson’s disease cellular models, cytotoxicity towards a set of human prostate cancer cell lines, and the antiradical effects towards DPPH. Scientific novelty derives from the isolation of cryptoechinuline B enantiomers, as well as from the biological effects observed in cell-based assays. While the experiments appear to have been properly conducted, authors should attempt to properly present the results through a more logical and “reader-friendly” way. As detailed below, while there is no necessity of conceptual changes or even additional experimental data, a series of issues should be addressed by the authors.
First, authors are advised to have the manuscript revised by an English editing service since grammar and language style is often rudimental as exemplified in:
Lines 43-44: “Another big (wide) group of metabolites which are commonly produced by …”.
Lines 45-46: “… only cryptoechinulines B and D have been reported as a results of echinulin-related indolediketopiperazines and auroglaucin-related polyketides pathway mixing”.
The sentence is unclear, authors being advised to consider: “Cryptoechinulines B and D derive from a mixed polyketide-amino acid biosynthetic pathway has evidenced by the echinulin- and auroglaucin-derived moieties.”.
Line 86: Revise “Cryptoechunuline B” to “Cryptoechinuline B”.
Line 107: “…at concentrations of up to 100…”.
Line 143: “…significant increase of on the viability…”.
Additional examples can be found throughout the entire manuscript.
Authors are also advised to use more indicative and suitable keywords, as “NMR data” or “cytotoxic activity” are unspecific.
Solely corresponding to my own opinion, since the NMR data has been provided (including 2D data) it would be relevant (but not mandatory) to further discuss the structural elucidation of the planar structure of cryptoechinuline B. While the NMR data is consistent with the planar structure, Table 2 should be revised, specifically concerning the carbon type (C16, C17, C18, C19 and C20). Based on the spectral data and the presented structures, the absolute configuration of the enantiomers appears to be incorrect, as the configuration of C26 is 26R and 26S, in 1a and 1b, respectively.
In order to provide a comprehensive view on the SAR of the compounds, it would be more convenient to jointly present the results on the neuroprotective effects in a unique subsection, instead of 2.2.1, 2.2.2 and 2.2.3, as the isolated compounds are structurally related.
As docetaxel was used as a reference drug (section 4.7.2), authors should provide the IC50 values, in order to unequivocally classify the observed cytotoxic effects of the echinulin derivatives towards prostate cancer cells as relevant, instead of an ambiguous classification of “moderate” (line 150).
Finally, the authors are also advised to consider the Instructions for Authors as several references were included in disagreement with it.
Author Response
Dear reviewer, thank you for the careful analysis of our manuscript. Below you can find the answers for your comments.
Q: Lines 43-44: “Another big wide group of metabolites which are commonly produced by …”.
A: It has been revised
Q: Lines 45-46: “… only cryptoechinulines B and D have been reported as a results of echinulin-related indolediketopiperazines and auroglaucin-related polyketides pathway mixing”.
The sentence is unclear, authors being advised to consider: “Cryptoechinulines B and D derive from a mixed polyketide-amino acid biosynthetic pathway has evidenced by the echinulin- and auroglaucin-derived moieties.”.
A: It has been revised
Q: Line 86: Revise “Cryptoechunuline B” to “Cryptoechinuline B”.
A: It has been revised
Q: Line 107: “…at concentrations of up to 100…”.
A: It has been revised
Q: Line 143: “…significant increase of on the viability…”.
A: “increase of the viability” is right variant.
Q: Authors are also advised to use more indicative and suitable keywords, as “NMR data” or “cytotoxic activity” are unspecific.
A: Several additional keywords have been added. We think that keywords “NMR data” and “cytotoxic activity” are often used in search request, therefore they should be kept here.
Q: While the NMR data is consistent with the planar structure, Table 2 should be revised, specifically concerning the carbon type (C16, C17, C18, C19 and C20).
A: Thank you for this notice. The table was revised.
Q: Based on the spectral data and the presented structures, the absolute configuration of the enantiomers appears to be incorrect, as the configuration of C26 is 26R and 26S, in 1a and 1b, respectively.
A: Thank you for the notice. Due to the error in the literature data (ref. 5, Gao H. et al. // Arch. Pharm. Res. 2013. V. 36, No. 8. P. 952-956) some mistakes were made in our manuscript too. Therefore, we have revised following points:
figure 1, structures of 1a and 1b, the absolute configurations were assigned for 1a and 1b as 12R, 26S, 29R and 12S, 26R, 29S, respectively (instead of 12R, 26S, 29S and 12S, 26R, 29R, respectively), the figure 2 caption was revisedQ: In order to provide a comprehensive view on the SAR of the compounds, it would be more convenient to jointly present the results on the neuroprotective effects in a unique subsection, instead of 2.2.1, 2.2.2 and 2.2.3, as the isolated compounds are structurally related.
A: Subsections 2.2.1, 2.2.2 and 2.2.3 have been joined.
Q: As docetaxel was used as a reference drug (section 4.7.2), authors should provide the IC50 values, in order to unequivocally classify the observed cytotoxic effects of the echinulin derivatives towards prostate cancer cells as relevant, instead of an ambiguous classification of “moderate” (line 150).
A: The information about docetaxel activity have been added in Table S1 (Supplementary data). The estimation of cytotoxicity as “strong”, “moderate” and “weak” often uses in many research papers in addition to exact values of IC50.
Q: Finally, the authors are also advised to consider the Instructions for Authors as several references were included in disagreement with it.
A: The references have been revised.